# Advancing Differential Privacy through Synthetic Dataset Alignment

## Abstract

Privacy in training data is crucial to protect sensitive personal information, prevent data misuse, and ensure compliance with legal regulations, all while maintaining trust and safeguarding individuals' rights in the development of ML models. Unfortunately, state-of-the-art methods that train ML models on image datasets with differential privacy constraints typically result in reduced accuracy due to noise. Alternatively, using synthetic data avoids the direct use of private data, preserving privacy, but suffers from domain discrepancies when compared to test data. This paper proposes a new methodology that combines both approaches by generating differentially private synthetic data closely aligned with the target domain, thereby improving the utility-privacy trade-off.

Our approach begins with creating a synthetic base dataset using a class-conditional generative model. To address the domain gap between the synthetic dataset and the private dataset, we introduce the **Privacy-Aware Synthetic Dataset Alignment (PASDA)**, which leverages the feature statistics of the private dataset to guide the domain alignment process. PASDA produces a synthetic dataset that guarantees privacy while remaining highly functional for downstream training tasks. Building on this, we achieve state-of-the-art performance, surpassing the most competitive baseline by over 13% on CIFAR-10. Furthermore, our $(1, 10^{-5})$-DP synthetic data achieves model performance on par with or surpassing models trained on the original STL-10, ImageNette and CelebA dataset. With zero-shot generation, our method does not require resource-intensive retraining, offering a synthetic data generation solution that introduces **privacy** to a machine learning pipeline with both high **efficiency** and **efficacy**.

## 1 Introduction

The rapid deployment and influence of AI brings the urgency of privacy and security. In traditional machine learning pipelines, private datasets used for model training are susceptible to various privacy breaches if adequate protections are not implemented. As illustrated in Figure (1) (a), the non-private pipeline exposes the trained classifier to reconstruction attacks and membership inference attacks, which can result in data leakage and the exposure of sensitive information. These attacks exploit the model's learned parameters to reconstruct private training data or determine the inclusion of specific data points, highlighting the critical need for strong privacy-preserving mechanisms to mitigate such risks and protect sensitive training data. With growing concerns around data privacy and security, the advancement of differential privacy (DP) (Dwork et al. (2014)) has become increasingly important in machine learning. Differentially Private Stochastic Gradient Descent (DPSGD) (Abadi et al. (2016a)) is a widely used method which ensures that individual data points remain confidential during model training. However, a significant drawback of DPSGD is the decrease in model accuracy due to the noise added to gradients in the model parameters.

While generating synthetic data is a promising alternative for privacy preservation, it is not always a fully effective solution. Synthetic data generation methods may struggle to capture the intricate distributions and specific nuances of the private dataset, potentially limiting their effectiveness, especially in scenarios requiring high fidelity (Fan et al. (2024); He et al. (2023)). Additionally, generative models might fail to represent certain classes or rare data patterns present in the private dataset, thereby compromising the utility of the synthetic data for downstream tasks. Recent

Figure 1: **Overview of PASDA** for generating differentially private synthetic datasets. To (a) protect privacy of private dataset during model training, we propose (b) PASDA which aligns substitute synthetic training data via differentially private statistics of the private dataset. Our results (c) shows that PASDA achieves superior performance compared to models trained on private datasets.

advances on this field includes training or finetuning generative models with differentially private techniques (Ghalebikesabi et al. (2023); Cao et al. (2021); Torkzadehmahani et al. (2019); Ho et al. (2021)), or use private dataset to provide guidance on generating process (Lin et al.). However, these methods often require substantial computational resources for training large generative models or iteratively generating large volumes of data, limiting their applicability in resource-constrained environments. Furthermore, most of these techniques have been primarily tested on low-resolution datasets, such as CIFAR-10 (32x32) (Krizhevsky et al. (2009)) or CelebA (64x64) (Liu et al. (2015)), restricting their use in more realistic applications.

In this work, we introduce **privatePrivacy-Aware Synthetic Dataset Alignment (pasda)**, a simple yet effective two-step paradigm, outlined in Figure (1) (b), to generate private and in-distribution synthetic data. First, we generate a fully synthetic dataset without any access to the private dataset, using pretrained class conditional generative models such as stable diffusion (Rombach et al. (2022)). Second, we align the distribution of the synthetic dataset with that of the private dataset while preserving privacy. This is achieved by extracting feature statistics from the private dataset with Gaussian mechanism Dwork et al. (2006), ensuring differential privacy. These statistics are then used to adjust the distribution of the synthetic dataset, producing high-quality synthetic images that closely match the target distribution. Finally, we can build downstream models upon this in-distribution synthetic dataset, which is guaranteed to be differentially private.

In conclusion, PASDA offers the following advantages:

- Privacy: Our approach requires minimal access to the private dataset, utilizing only feature statistics to guide the generation of synthetic data. This method performs well for image classification tasks, even under strong privacy constraints.

- Efficiency: By leveraging pretrained models for dataset generation, PASDA generates synthetic data in a zero-shot fashion, significantly reducing computational costs, even in case of high resolution dataset.

- Effecacy: When training downstream models on our synthetic datasets with $(1, 10^{-5})$-DP guarantees, we established new SOTA on CIFAR-10, and achieved performance that is on par with or surpasses models trained on the private data across various datasets such as STL-10 (Coates et al. (2011)), ImageNette (Howard), and CelebA (Liu et al. (2015)).

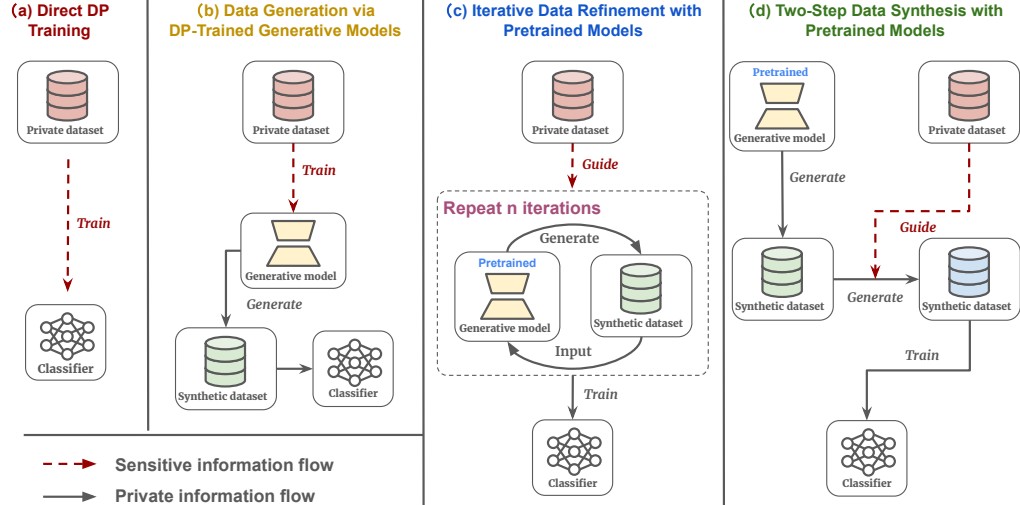

Figure 2: **Comparison with existing methods.** (a) Directly training the classifier on private data using DP techniques. (b) Training a DP generative model on private data to generate images for downstream training. (c) Using private data to guide a pretrained generative model, and iteratively using the generated dataset as the input to generate more training data. (d) Our method aligns synthetic data with private data through one-time DP access to the private data.

## 2 RELATED WORKS

### 2.1 SYNTHETIC DATA FOR COMPUTER VISION

Recent research on harnessing synthetic data to enhance computer vision systems has shown notable advancements, particularly in the realm of image recognition. Gowal et al. (2021) demonstrated that even low-quality synthetic data could substantially bolster neural network robustness against adversarial attacks. In parallel, Li et al. (2022) introduced an innovative approach using BigGAN and VQGAN to create a synthetic, pixel-wise annotated ImageNet dataset, which significantly streamlines the training process for segmentation models. Azizi et al. (2023) further showed that fine-tuning class-conditional generative diffusion models on ImageNet enhances classification accuracy through the use of photorealistic synthetic samples. Complementing these findings, He et al. (2023) confirmed that synthetic data could indeed enhance model robustness, while Sarıyıldız et al. (2023) underscored the utility of synthetic ImageNet datasets under specific conditions, contributing to the dialogue on the capabilities and constraints of synthetic data in image recognition. Further more, significant work in areas like object detection and semantic segmentation has also been pursued. Lin et al. (2023) highlighted that synthetic images could remarkably improve few-shot object detection, and Li et al. (2021) demonstrated the efficacy of a GAN-based network in boosting semantic segmentation performance across various applications.

### 2.2 DIFFERENTIALLY PRIVACY AND SYNTHETIC DATA GENERATION

In the evolving field of differentially private synthetic data generation, several pioneering methods have been developed, enhancing data privacy across various applications. Building on the foundation of existing methods in the field, we categorize the approaches, including our work, into four distinct types, each addressing differential privacy and data utility in different ways, as shown in Figure (2).

**Direct DP Training** The first category includes methods like DPSGD proposed by Abadi et al. (2016b); De et al. (2022), which directly train classifiers on private data using differential privacy techniques. While this ensures privacy, it often results in significant reduction of model performance due to the added noise required by DP constraints, especially when the privacy budget is tight.

**Data Generation via DP-Trained Generative Models** The second category includes most of the DP-guaranteed synthetic data generation approaches, which train generative models on private data with DP to generate synthetic datasets for downstream training. The key challenge here is to adapt these models to generate both domain-aligned and high-quality synthetic dataset with limited private data. Ghalebikesabi et al. (2023), Dockhorn et al. (2023) and Lyu et al. proposed Diffusion models

that integrate differential privacy into the generation of synthetic images. Li et al. (2024) employs semantic-aware pretraining for diffusion models, allowing efficient generation of differentially private synthetic images by leveraging public and private datasets. Wang et al. (2024a) leverage noise addition in the initial forward process steps to save privacy budget during the training of diffusion model. Most recently, Tsai et al. (2024) proposes a parameter-efficient fine-tuning strategy for diffusion models under differential privacy constraints, achieving state-of-the-art results by reducing the number of trainable parameters with LoRA modules while balancing the privacy-utility trade-off.

**Iterative Data Refinement with Pretrained Models** The third category encompasses image generation with pretrained model and iterative guidance, such DPSDA ( Lin et al.). These methods iteratively use the synthetic data to generate additional training data, gradually aligning the synthetic data with the real data. While this improves data alignment and does not need training, it suffers from significant computation cost and repeated access to the private dataset, increasing privacy risks.

**Two-Step Data Synthesis with Pretrained Models** Finally, PASDA introduces a new synthetic data generation paradigm consisting of two steps: (1) generate a synthetic base dataset, and (2) align the distribution of this base dataset with the private dataset. In contrast to previous methods, which often require substantial computational resources for training large generative models or generate a large volume of data, PASDA is designed to be more efficient and effective at the same time. By leveraging pretrained models for inference without retraining, PASDA minimizes the need for extensive computational power, making it more suitable for resource-constrained environments. Additionally, while many existing methods are primarily tested on low-resolution datasets such as CIFAR-10 (32x32) Krizhevsky et al. (2009) or CelebA (64x64) Liu et al. (2015), our approach can well handle higher-resolution datasets. This allows PASDA to be applied in more demanding applications where higher-quality synthetic data is required, without compromising on privacy or utility.

## 3 METHODOLOGIES

### 3.1 PRELIMINARIES

In recent years, there has been a significant focus on enhancing the privacy and utility of machine learning models through various techniques. DP (Dwork et al. (2006)) has emerged as a key framework for ensuring that the inclusion or exclusion of a single data point does not significantly affect the outcome of any analysis, thus preserving the privacy of individuals in the dataset.

**Differential Privacy** A randomized mechanism $M : D \rightarrow \mathcal{R}$ with domain $D$ and range $\mathcal{R}$ satisfies $(\epsilon, \delta)$-differential privacy if, for any two adjacent datasets that differ on a single element $D, D' \in D$ and for any subset of outputs $S \subseteq \mathcal{R}$, it holds that: $Pr[M(D) \in S] \leq e^\epsilon Pr[M(D') \in S] + \delta$, where $\epsilon$ and $\delta$ are non-negative parameters controlling the privacy loss, and $Pr$ refers to a probability measure. DP forms the foundation for various privacy-preserving mechanisms, including the Gaussian Mechanism Dwork et al. (2006). Expanding on these principles, DP Stochastic Gradient Descent (DPSGD) Abadi et al. (2016b) has been introduced as a privacy-preserving optimization algorithm. It integrates noise into the gradient descent process, allowing the training of machine learning models while ensuring data privacy. Complementing DPSGD, the Moments Accountant technique has been proposed for better privacy loss tracking in the DPSGD algorithm Abadi et al. (2016b). This method enhances the privacy analysis, offering more accurate privacy loss estimates and effective privacy budget management. DPSGD has been incorporated into the optimization process of many deep learning approaches to ensure privacy.

**Rényi Differential Privacy** Rényi Differential Privacy (RDP, Mironov (2017)) is an extension of differential privacy that provides a more flexible framework for analyzing and tracking the privacy loss over multiple computations.

**Definition 1** (Rényi Differential Privacy). *A randomized mechanism $M$ satisfies $(\alpha, \epsilon)$-Rényi differential privacy if for all adjacent datasets $D$ and $D'$ it holds that: $D_\alpha(M(D)\|M(D')) \leq \epsilon$, where $D_\alpha$ is the Rényi divergence of order $\alpha$ between the distributions of $M(D)$ and $M(D')$.*

The Gaussian mechanism can also be analyzed under the RDP framework, providing a tighter bound on the privacy loss.

**Theorem 1** (RDP of the Gaussian Mechanism(Mironov (2017)). *For the Gaussian mechanism with noise $\mathcal{N}(0, \sigma^2)$, the RDP parameter $\epsilon(\alpha)$ is given by: $\epsilon(\alpha) = \frac{\alpha s^2}{2\sigma^2}$, where $s$ is the $\ell_2$-sensitivity of the query function.*

Figure 3: **Framework of PASDA** for generating differentially private synthetic datasets. In the first step, we leverage a generative model to populate synthetic samples. In the second step, we privately align the synthetic dataset with the private dataset by privately augmenting the embeddings of synthetic data and decode the embeddings back to image space.

Here, the defition of the $\ell_2$-sensitivity is given by:

**Definition 2** (L2 Sensitivity). *The L2 sensitivity of a function $f : D^n \to \mathbb{R}^k$ is the maximum change in the function's output, measured by the Euclidean distance, when a single entry in the input dataset is modified. Formally, for two neighboring datasets $D$ and $D'$ that differ by at most one element, the L2 sensitivity is defined as: $\Delta_2 f = \max_{D,D'} \|f(D) - f(D')\|_2$, where $\| \cdot \|_2$ represents the L2 norm (Euclidean distance) between the function's outputs.*

RDP provides a powerful tool for privacy analysis in iterative algorithms like DPSGD, allowing for more accurate composition and tighter privacy guarantees. It has been widely adopted in privacy-preserving machine learning to improve the utility of models trained under differential privacy constraints. RDP can be converted to standard DP easily with the following lemma:

**Lemma 1** (RDP to DP Conversion. (Mironov (2017))). *If a randomized mechanism $\mathcal{M}$ guarantees $(\alpha, \epsilon)$-RDP $(\alpha > 1)$, then it also obeys $(\epsilon + \log(1/\delta)/(\alpha - 1), \delta)$-DP.*

### 3.2 PRIVATEPRIVACY-AWARE SYNTHETIC DATASET ALIGNMENT

This paper addresses the challenge of training machine learning models on image datasets under differential privacy constraints by leveraging a pretrained foundation model. Our methodology involves several key steps: generating a synthetic dataset using class-conditional generative model, aligning the synthetic dataset with the private dataset's distribution using privatePrivacy-Aware Synthetic Dataset Alignment (pasda), and generating the final synthetic dataset with the unCLIP model Rombach et al. (2022). Below, we detail each step of our approach.

**Base Dataset Generation**  We begin by generating a fully private synthetic dataset using class-conditional generative model, e.g., stable diffusion (Rombach et al. (2022)) in our method, as shown in Figure (3) Step 1. Stable diffusion is a powerful generative model capable of producing high-quality images from text. We use category-specific prompts in the format of "a photo of a {category}" to generate synthetic images. This process ensures that the generated data is entirely synthetic and does not involve direct access to the private dataset, thus preserving privacy by design.

**Domain Alignment with Gap Embedding**  The synthetic data generated using generative models may have a different distribution compared to the private dataset, which can adversely affect model performance. To address this, we use the domain gap vector in CLIP embedding space(Wang et al. (2024b)) to describe the difference between the synthetic and private dataset, and use that to align their distributions.

We illustrate our PASDA process by applying it to a single category, as shown in **Figure (3)**. For each category, we first utilize CLIP (Radford et al. (2021)) to obtain embeddings for both the private

---

**Algorithm 1** DP-guaranteed Domain Alignment for Synthetic and Private Dataset

---

1: **Input:** Synthetic dataset from class-conditional generative model $\mathbb{D}^{(\text{syn})}$,
2:       Private dataset of real images $\mathbb{D}^{(priv)}$,
3:       Pretrained CLIP$(\cdot)$(Radford et al. (2021) and unCLIP$(\cdot)$ (Ramesh et al.),
4:       Maximum norm for clipping private embeddings$\kappa$,
5:       Number of clusters $K$,
6:       Privacy budget $(\epsilon, \delta)$.
7: **Output:** Synthetic dataset $\mathbb{D}^{(\text{pasda})}$.
8: Initialize $\mathbb{D}^{(\text{pasda})} \leftarrow \emptyset$
9: **for** each category $c$ **do**
10:     $\mathbb{V}_c^{(priv)} \leftarrow \text{CLIP}(\mathbb{D}_c^{(priv)}), \mathbb{V}_c^{(\text{syn})} \leftarrow \text{CLIP}(\mathbb{D}_c^{(\text{syn})})$                // Extract CLIP embeddings
11:     $(\mathcal{S}^{(\text{syn})}, \mathcal{S}^{(priv)}) \leftarrow \text{ClusterMatch}(\mathbb{V}_c^{(\text{syn})}, \mathbb{V}_c^{(priv)}, K)$    // Spectral clustering and Hungarian matching
12:     **for** $m = 1$ to $K$ **do**
13:         $\Delta_m \leftarrow \text{DiffPrivMean}(\mathcal{S}_m^{(priv)}, \epsilon, \delta, \kappa) - \text{Mean}(\mathcal{S}_m^{(\text{syn})})$   // Privacy-aware Domain gap estimation ,
    see Algorithm (2)
14:         **for** $\boldsymbol{v}^{(\text{syn})} \in \mathcal{C}_m^{(\text{syn})}$ **do**
15:             $\mathbb{D}^{(\text{pasda})} \leftarrow \mathbb{D}^{(\text{pasda})} \cup \text{unCLIP}(\boldsymbol{v}^{(\text{syn})} + \Delta_m)\}$         // Image generation from embeddings
16:         **end for**
17:     **end for**
18: **end for**
19: **return** Synthetic dataset $\mathbb{D}^{(\text{pasda})}$

---

and synthetic datasets, capturing the semantic content of images in a high-dimensional latent space. To account for the semantic diversity and intra-class variations, we avoid treating each class as a single, isolated semantic entity. Instead, we partition each class into $K$ clusters using a clustering algorithm, such as spectral clustering. We then apply the Hungarian matching algorithm to pair corresponding clusters from the synthetic and private datasets by minimizing the Euclidean distance between cluster centroids, ensuring matched clusters share similar semantics. Next, we compute the domain gap for each matched pair between the private and synthetic datasets. The domain gap is the difference in the distributions of their CLIP embeddings. We calculate this gap using the expected differences of all pairs between the source (private) and target (synthetic) datasets, which is mathematically equivalent to the difference in the means of their embeddings. The domain gap vector $\boldsymbol{\Delta}v$ is given by $\mathbb{E}[\boldsymbol{v}^{(priv)}] - \mathbb{E}[\boldsymbol{v}^{(\text{syn})}]$ (Line (13) in Algorithm (1)), where $\boldsymbol{v}^{(priv)}$ and $\boldsymbol{v}^{(\text{syn})}$ are the CLIP embeddings of the private and synthetic datasets, respectively.

**Differential Private domain gap with Gaussian Mechanism**     To ensure differential privacy, we add Gaussian noise to the domain gap vector in Algorithm (2), shown in Figure (3) Step 2. This step is crucial to maintain privacy while aligning the distributions. The differentially private domain gap vector is computed as $\hat{\boldsymbol{\Delta}}v = \boldsymbol{\Delta}v + \mathcal{N}(0, \sigma^2)$ (Line (13) in Algorithm (1)), where $\mathcal{N}(0, \sigma^2)$ represents Gaussian noise with mean 0 and variance $\sigma^2$. The noise addition follows the Gaussian mechanism, which is defined to satisfy RDP. Specifically, the Gaussian mechanism adds noise calibrated to the sensitivity of the function and the desired privacy parameters. The noise scale $\sigma$ is then given as Table (1):

**Corollary 1** (Noise multiplier calculation). *Combining Theorem (1) and Lemma (1), the formula for the noise multiplier with $(\epsilon, \delta)$-DP is given by:* $\sigma = s\sqrt{\min_{\alpha > 1 + \frac{\log(1/\delta)}{\epsilon}} \left( \frac{\alpha}{2\left(\epsilon - \frac{\log(1/\delta)}{\alpha - 1}\right)} \right)}$.

where $\epsilon$ and $\delta$ are the privacy parameters. This ensures that the privacy loss is controlled, and the added noise is sufficient to protect individual data points' privacy in the dataset.

We adjust the CLIP embeddings of the synthetic data by adding the differentially private domain gap vector: $\boldsymbol{v}_{\text{adjusted}} = \boldsymbol{v}_{\text{synthetic}} + \hat{\boldsymbol{\Delta}}v$, This adjustment ensures that the synthetic data's distribution is more closely aligned with the private dataset's distribution while preserving differential privacy.

**Converting Embeddings to Images with unCLIP**     After adjusting the synthetic data embeddings, we use the unCLIP model to generate the final synthetic dataset. To preserve category-specific semantics and mitigate the impact of noise, we guide the unCLIP generation process with category cues to ensure that the generated images are semantically consistent with their intended categories.

---

**Algorithm 2** *DiffPrivMean*: Differentially Private Mean Calculation

---

1: **Input:** Embedding set from private data $\mathbb{V}^{(priv)}$,
2:   Privacy budget in form of $(\epsilon, \delta)$,
3:   Maximum norm for clipping private embeddings $\kappa$.
4: **Output:** Differentially private mean estimation of the given dataset $\bar{\boldsymbol{v}}^{(priv)}$

5: $\boldsymbol{v}_{sum}^{(priv)} \leftarrow 0$
6: **for** $\boldsymbol{v}^{(priv)} \in \mathbb{V}^{(priv)}$ **do**
7:  $\hat{\boldsymbol{v}}^{priv} \leftarrow \boldsymbol{v}^{(priv)} \cdot \min\left(1, \frac{\kappa}{\|\boldsymbol{v}^{(priv)}\|_2}\right)$      // Norm clipping using predefined threshold $\kappa$
8:  $\boldsymbol{v}_{sum}^{(priv)} \leftarrow \boldsymbol{v}_{sum}^{(priv)} + \hat{\boldsymbol{v}}^{(priv)}$
9: **end for**
10: $\sigma \leftarrow \min_{\alpha > 1 + \frac{\log(1/\delta)}{\epsilon_\delta}} \left(\frac{\alpha s^2}{2\left(\epsilon_\delta - \frac{\log(1/\delta)}{\alpha - 1}\right)}\right)$   // Calculate the noise multiplier, see Corollary (1)
11: $\bar{\boldsymbol{v}}^{(priv)} \leftarrow (\boldsymbol{v}_{sum}^{(priv)} + \mathcal{N}(0, \sigma^2 \kappa^2 \boldsymbol{I})) / \|\mathbb{V}^{(priv)}\|_2$  // Add noise to preserve differential privacy
12: **return** $\bar{\boldsymbol{v}}^{(priv)}$

---

# 4 EXPERIMENTS

## 4.1 EXPERIMENTAL SETUP

**Datasets** We used four datasets in our experiments: CIFAR-10 (Krizhevsky et al. (2009)) STL-10 (Coates et al. (2011)), ImageNette (Howard) and CelebA (Liu et al. (2015)). CIFAR-10, STL-10, and ImageNette contain 10 classes, while CelebA is used for binary gender classification.

CIFAR-10 is a widely-used benchmark for visual classification and privacy research, given its low resolution ($32 \times 32$) and large dataset size ($5,000$ images per class). However, real-world applications often present greater complexity. To extend our evaluation, we include ImageNette and STL-10, which offer higher resolution images ($160 \times 160$ and $96 \times 96$, respectively), with STL-10 being more challenging due to its smaller dataset size (500 images per class).

To further assess the effectiveness of our method across diverse data distribution, we also utilize CelebA, a dataset for facial attribute classification, which introduces a more complex and varied set of images. To reduce computational overhead, we randomly select a subset of 5,000 images from CelebA for our evaluations, rather than using the full dataset.

**Models** We evaluated the approaches by the performance of downstream classification using two popular neural network models commonly adopted in privacy literature: ConvNet (Krizhevsky et al. (2012)) and ResNet-9 (He et al. (2016)), as well as two deeper models for higher resolution tasks ResNet-50 (He et al. (2016)) and VGG-11 (Simonyan & Zisserman (2015)). ConvNet is a simple convolutional network without a BatchNorm layer, allows for direct comparison with the baseline DPSGD method. However, for ResNet-50 and VGG-11, which contain BatchNorm layers, DPSGD cannot fully guarantee privacy due to potential leakage of data through BatchNorm statistics. Therefore, we did not implement DPSGD on these two models.

**Baselines** We compared our methods with most recently proposed DP synthetic generation methods: DPDM (Dockhorn et al. (2023)), PrivImage (Li et al. (2024)) and DPSDA (Lin et al.), as well as one baseline with direct class-conditional text-to-image generation with stable diffusion v2 (SD-v2) (Rombach et al. (2022)). For comparison, we also implement DPSGD (Abadi et al. (2016b)) as a baseline to demonstrate the performance of direct training on private dataset with a given privacy budget.

**Hyperparameters** Our method involves five key hyperparameters: the privacy budget $(\epsilon, \delta)$, the maximum norm for CLIP embeddings $\kappa$, the number of clusters $K$, and the reduced dimensionality $d$ for the clustering algorithm. For all evaluations, we fix $\kappa = 20$. The selection of $\kappa$ is informed by the observation that the majority of images in the tested dataset have embedding norms approximately around 20. Regarding the number of clusters, we set $K = 1$ for STL-10 and $K = 10$ for CIFAR-10, CelebA, and ImageNette. This choice is motivated by the number of images per class: STL-10 has fewer images per class (500), while CIFAR-10, ImageNette, and CelebA have significantly more (5000 for CIFAR-10, $\sim 9470$ for ImageNette, and 2500 for CelebA). Although more clusters can help preserve diversity, they also reduce the number of images in each cluster, making the results more susceptible to noise. We set $d = 10$ as the default for all evaluations. A detailed analysis of

the hyperparameters can be found in section 4.3, and the hyperparameters for other baselines are provided in Appendix B.

For the downstream task, we trained ConvNet, ResNet-50 and VGG-11 with a batch size of $128$, a learning rate of $10^{-2}$, weight decay of $5 \times 10^{-4}$, and a momentum of $0.9$ with the SGD optimizer. For ResNet-9, we used a batch size of $64$, a learning rate of $5 \times 10^{-2}$, weight decay of $10^{-3}$, and a momentum of $0.9$.

## 4.2 MAIN RESULTS

### 4.2.1 COMPARISON WITH BASELINE METHODS

The comparison with other baseline methods is shown in Table (1). For each method, we generate $50,000$ of synthetic images to train downstream classifiers for fair comparison. We show that PASDA achieves the **best performance** across both datasets and model architectures, marking the **new SOTA** on these tasks. Moreover, we find that STL-10 poses a significantly more challenging task for all methods due to its higher resolution and smaller dataset size compared to CIFAR-10. Recently introduced diffusion model retraining methods, such as PrivImage (Li et al. (2024))

| Dataset | Method | Architectures | |
|---------|--------|:-------------:|:--------:|
| | | ConvNet | ResNet-9 |
| STL-10 | DP-LDM (Lyu et al.) | - | - |
| | DPDM (Dockhorn et al. (2023)) | 10.2 | 9.8 |
| | DPSDA (Lin et al.) | 19.8 | 24.5 |
| | PrivImage (Li et al. (2024)) | 11.3 | 10.4 |
| | SD-v2 (Rombach et al. (2022)) | 54.8 | 59.8 |
| | PASDA (ours) | **59.5** | **68.2** |
| CIFAR-10 | DP-LDM (Lyu et al.) | - | 51.3 |
| | DPDM (Dockhorn et al. (2023)) | - | 14.7 |
| | DPSDA (Lin et al.) | 25.7 | 47.1 |
| | PrivImage (Li et al. (2024)) | 33.2 | 31.7* |
| | SD-v2 (Rombach et al. (2022)) | 52.4 | 56.7 |
| | PASDA (ours) | **62.0** | **70.3** |

Table 1: Performance Comparison on STL-10 and CIFAR-10 Datasets. *The original paper reported a PrivImage performance of 66.2 on CIFAR-10 using ResNet-9. The results shown in the table were obtained using the published code with its default configuration.

and DPDM (Dockhorn et al. (2023)), perform poorly on STL-10, likely due to the difficulties stemming from its high resolution and limited sample size. Surprisingly, the pretrained text-to-image model, SD-v2, outperforms all other baselines. We attribute this success to SD-v2's strong generative capabilities and the similarity between its pretraining dataset and both STL-10 and CIFAR-10. Consequently, SD-v2 performs consistently well across different resolutions on both datasets, while other models struggle with generating high-resolution synthetic images with limited data.

### 4.2.2 COMPARISON WITH THE ORIGINAL DATASET

We further evaluate PASDA on STL-10, ImageNette, and CelebA using various network architectures with synthetic datasets that are ten times the original datasets. Surprisingly, the performance of models trained on datasets generated by PASDA is comparable to or better than those trained on the original data, as shown in Table (2). Notably, the accuracy is improved by 2.2% and 10.6% on STL-10 and ImageNette using ConvNet, respectively. For ResNet-50 and VGG11, the performance of models trained using data generated by PASDA is slightly lower than that of models trained on the original datasets, but still comparable. The results indicate that PASDA provides a practical solution to replace private datasets with synthetic datasets, ensuring strong privacy protection while maintaining or even surpassing the performance with the original data.

### 4.2.3 VISUALIZATION OF SYNTHETIC IMAGES

We visualize synthetic images generated by PrivImage, DPSDA, SD-v2, and our method under a privacy budget of $(1, 10^{-5})$ Figure (4). PrivImage struggles to produce semantically clear images under this budget. DPSDA generates high-quality images, though some lack semantic consistency (e.g., certain airplane and dog images). This inconsistency may result from the high noise introduced during image selection. While SD-v2 produces high-quality images, its style diverges from CIFAR-10, favoring a more photographic aesthetic. In contrast, PASDA generates semantically consistent, high-quality images with a style closely aligned to CIFAR-10. Notably, in the ship category, SD-v2 images exhibit a vintage tone, whereas PASDA produces more natural tones, likely guided by the CIFAR-10 dataset. Overall, PASDA delivers in-distribution, semantically accurate images, contributing to its strong downstream performance. See appendix C for more visualizations.

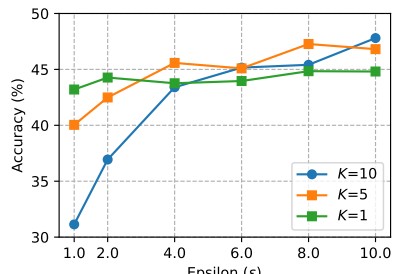

Figure 4: Comparison of original CIFAR-10 images (leftmost column) with synthetic images generated by various approaches (the four columns to the right).

| Dataset | Method | Architectures | | |
| --- | --- | --- | --- | --- |
| | | ConvNet | ResNet-50 | VGG-11 |
| **STL-10** | Private Data | 57.3 | 64.8 | 64.0 |
| | DPSGD | 29.0 | – | – |
| | PASDA (ours) | **59.5** | **63.0** | **59.5** |
| **ImageNette** | Private Data | 51.6 | 70.6 | 71.4 |
| | DPSGD | 24.4 | – | – |
| | PASDA (ours) | **62.2** | **67.2** | **60.7** |
| **CelebA** | Private Data | 94.0 | 90.2 | 92.6 |
| | DPSGD | 58.1 | – | – |
| | PASDA (ours) | **89.3** | **83.2** | **86.3** |

Table 2: Comparison of model accuracy trained on the private dataset versus PASDA-generated datasets across different architectures on STL-10, CelebA, and ImageNette. PASDA operates under a privacy constraint of $(1, 10^{-5})$.

Figure 5: Performance comparison across different values of the privacy budget ($\epsilon$) for varying numbers of clusters ($K = 1, 5, 10$).

### 4.3 ABLATION STUDIES

#### 4.3.1 SAMPLE SIZE $N$

Figure (6) illustrates the classification accuracy of models trained on synthetic datasets generated by PASDA and SD-v2 across three datasets: STL-10, ImageNette, and CelebA, with varying sample size multipliers. As the sample size increases (from 1x to 10x), PASDA consistently outperforms SD-v2. In particular, PASDA achieves performance comparable to models trained on private datasets when the sample size is scaled to seven times the original size on STL-10, three times on ImageNette, and ten times on CelebA. For the STL-10 dataset (Figure (6)(a)), PASDA exceeds the baseline at higher sample multipliers, while in the ImageNette dataset (Figure (6)(b)), it even surpasses both SD-v2 and the baseline at only three times the original size. Although PASDA does not quite reach the baseline accuracy for CelebA, it approaches the baseline at 10x, demonstrating its capacity to maintain high performance while ensuring strong privacy guarantees.

#### 4.3.2 PRIVACY BUDGET $(\epsilon, \delta)$ AND NUMBER OF CLUSTERS $K$

We evaluate the performance of our proposed method across varying privacy budgets by adjusting the parameter $\epsilon$, as shown in Figure (5). For each configuration, we generate 5,000 samples and present the corresponding performance as a function of $\epsilon$ in the accompanying figure. When the number of clusters is set to $K = 10$, we observe a gradual improvement in performance as $\epsilon$ increases. In contrast, when $K = 1$, the performance remains relatively stable despite increases in $\epsilon$. This phenomenon can be attributed to the fact that, with $K = 1$, the method is restricted to accessing only the noisy mean vector for each category, thereby imposing a performance ceiling. To understand this bottleneck, consider the extreme case where no noise is added; in this scenario, our method relies solely on the mean vectors of each category. The performance remains constrained because the mean vectors encode limited information, thereby restricting the richness of insights that can be derived from the private dataset. When $K = 5$, the performance follows a trend that falls between that of $K = 1$ and $K = 10$, demonstrating intermediate behavior as expected.

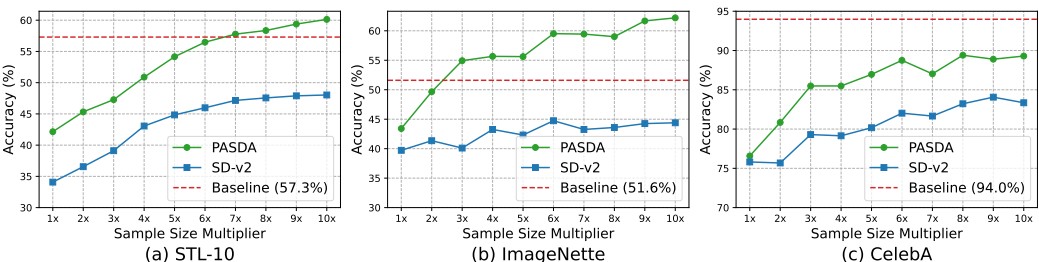

Figure 6: Accuracy of ConvNet as the number of generated samples scales on STL-10.

Increasing the number of clusters, $K$, allows the method to exploit a more diverse set of cluster centroids, providing richer insights into the private dataset. However, this enhancement comes at the cost of increased noise. Specifically, smaller cluster sizes amplify the impact of noise on the calculation of the mean vector, as the aggregate embedding sum decreases, thereby reducing the relative influence of the noise term (i.e., $\bar{\boldsymbol{v}}^{(priv)} \leftarrow (\boldsymbol{v}_{sum}^{(priv)} + \mathcal{N}(0, \sigma^2 \kappa^2 \boldsymbol{I}))/\|\mathbb{V}^{(priv)}\|_2$, see Line (11) in Algorithm (2)). Consequently, under conditions of high noise (i.e., lower $\epsilon$), a smaller number of clusters ($K = 1$) proves to be more effective than $K = 10$. Conversely, when the noise level is reduced (i.e., higher $\epsilon$), increasing the number of clusters leads to improved performance, as the greater diversity of cluster centroids provides more information about the private dataset.

## 5 DISCUSSION AND LIMITATIONS

**Privacy Concern on Pretraining Data** In the context of PASDA, a key privacy concern revolves around the pretraining data used by the foundation models, which PASDA relies on to generate training images. While PASDA itself provides strong differential privacy guarantees for the datasets it interacts with directly, it does not extend these protections to the pre-training data used in these foundation models. This is because PASDA has no control or visibility into that pre-training process. While this issue is beyond the scope of the PASDA approach, it cannot be ignored. The research community at large recognizes that pretraining data privacy is a critical issue that affects the entire lifecycle of the foundation models, not just the downstream tasks addressed by PASDA. Addressing this issue will require better privacy protection and auditing methods for foundation models during the pretraining phase to ensure the end-to-end privacy of the models involved.

**When Pretraining Data Fails to Cover Target Domains** PASDA is an innovative, training-free method that solely relies on pretrained foundation models to generate synthetic images. A concern arises when the target dataset's distribution is not represented in the foundation model's pretraining data. For instance, many foundation models are trained on large-scale image-text pairs collected from the Internet, but data from specialized fields may be underrepresented. This includes images from areas such as X-rays, MRI, CT scans, and cosmic images from astronomy. In such cases, PASDA may perform poorly because the generative model has not been exposed to the specific distribution it needs to generate on. Addressing the challenge of generating private, out-of-distribution data remains an interesting issue, which we plan to explore in future work.

## 6 CONCLUSIONS

In this paper, we introduce PASDA, a method that tackles the privacy-utility tradeoff by generating differentially private synthetic data tailored to the target domain. PASDA leverages pretrained class-conditional generative models and feature statistics from private datasets to minimize the domain gap while maintaining strong privacy guarantees. Our method excels in its privacy, efficiency, and efficacy. PASDA relies solely on feature statistics to guide the synthetic data generation process, which eliminates the need to train large models. This proposed paradigm greatly reducing computational costs, even for high-resolution datasets. PASDA established a SOTA on CIFAR-10 benchmark as compared to previous DP synthetic dataset generation approach. Moreover, our results across STL-10, ImageNette, and CelebA, demonstrate that models trained on PASDA-generated synthetic data perform on par with, and in some cases exceed, those trained on the original private data. In summary, PASDA marks a significant advance in privacy-preserving synthetic data generation, offering a practical and scalable solution for high-utility private machine learning applications.

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

## A  PROOF OF COROLLARY (1) (NOISE MULTIPLIER CALCULATION)

*Proof.* We begin by noting that, from Theorem (1), the Rényi Differential Privacy (RDP) parameter $\epsilon(\alpha)$ of the Gaussian mechanism with noise $\mathcal{N}(0, \sigma^2)$ is given by the equation

$$\epsilon(\alpha) = \frac{\alpha s^2}{2\sigma^2},$$

where $\alpha > 1$ is the order of the RDP, $s$ is the $\ell_2$-sensitivity of the query function, and $\sigma$ is the noise multiplier (i.e., the standard deviation of the added noise).

Next, we employ Lemma (1), which states that if a mechanism satisfies $(\alpha, \epsilon(\alpha))$-RDP, then it also satisfies $(\epsilon', \delta)$-Differential Privacy, where $\epsilon'$ is given by

$$\epsilon' = \epsilon(\alpha) + \frac{\log(1/\delta)}{\alpha - 1},$$

for any $\delta > 0$. Therefore, to satisfy $(\epsilon, \delta)$-DP, we require

$$\epsilon = \epsilon(\alpha) + \frac{\log(1/\delta)}{\alpha - 1}.$$

Substituting $\epsilon(\alpha) = \frac{\alpha s^2}{2\sigma^2}$ from Theorem (1) into this expression, we obtain the equation

$$\epsilon = \frac{\alpha s^2}{2\sigma^2} + \frac{\log(1/\delta)}{\alpha - 1}.$$

We now solve for $\sigma^2$ in terms of $\epsilon$, $s$, $\alpha$, and $\delta$. Rearranging the above equation, we get

$$\frac{\alpha s^2}{2\sigma^2} = \epsilon - \frac{\log(1/\delta)}{\alpha - 1},$$

which leads to

$$\sigma^2 = \frac{\alpha s^2}{2\left(\epsilon - \frac{\log(1/\delta)}{\alpha - 1}\right)}.$$

Thus, the noise variance $\sigma^2$ is determined by $\alpha$, $\epsilon$, $s$, and $\delta$.

To minimize the noise while ensuring $(\epsilon, \delta)$-DP, we seek to minimize $\sigma^2$ over all $\alpha > 1$. Specifically, we minimize the expression

$$\sigma^2 = s^2 \frac{\alpha}{2\left(\epsilon - \frac{\log(1/\delta)}{\alpha - 1}\right)}.$$

The optimal value of $\alpha$ must satisfy $\alpha > 1 + \frac{\log(1/\delta)}{\epsilon}$ to ensure that the denominator remains positive, as the term $\epsilon - \frac{\log(1/\delta)}{\alpha - 1}$ must be strictly positive for $\sigma^2$ to be well-defined.

Thus, the noise multiplier $\sigma$ is given by

$$\sigma = s \sqrt{\min_{\alpha > 1 + \frac{\log(1/\delta)}{\epsilon}} \frac{\alpha}{2\left(\epsilon - \frac{\log(1/\delta)}{\alpha - 1}\right)}}.$$

This completes the proof of Corollary (1). $\qquad\square$

## B  HYPERPARAMETERS FOR TRAINING BASELINE METHODS ON STL-10

In this work, we reproduce the results of DPSDA[1] (Lin et al.), DPDM[2] (Dockhorn et al. (2023)), and PrivImage[3] (Li et al. (2024)) for comparison. The models were trained on four NVIDIA A4500 GPUs.

### B.1  STL-10 (COATES ET AL. (2011))

#### B.1.1  DPSDA (LIN ET AL.)

We generated images using class-conditioned DPSDA with a pretrained improved diffusion model on ImageNet at a resolution of $64 \times 64$. The feature extractor was InceptionV3 (Szegedy et al. (2016)), with a count threshold of 10 and a lookahead degree of 1. The noise multiplier was computed as 15.83 under the privacy parameters $\epsilon = 1$ and $\delta = 10^{-5}$. The process was carried out over 20 iterations, generating 50,000 samples per iteration, with the degree of variation increasing linearly from 0 to 40 over the iterations.

#### B.1.2  PRIVIMAGE (LI ET AL. (2024))

**Semantic Query**  The semantic query classifier was trained using ResNet-50 (He et al. (2016)) with a batch size of 256, a learning rate of $10^{-2}$, and 60 epochs. The differential privacy parameters were set to $\epsilon = 0.01$ and $\delta = 10^{-5}$.

**Pretraining**  The Noise Conditional Score Network (NCSN++)**?** was trained using the Elucidated Diffusion Models (EDM) frameworkKarras et al. (2022) on the ImageNet dataset at a resolution of $96 \times 96$, with an exponential moving average (EMA) rate of 0.999. The model architecture included attention at a resolution of 16 and channel multipliers of $[1, 2, 4]$. Optimization was performed using the Adam optimizer with a learning rate of $1 \times 10^{-4}$ and no weight decay. The deterministic DDIM sampler with 50 steps was used, with a time range from $t_{\min} = 0.002$ to $t_{\max} = 80$, $\rho = 7$, and no guidance scaling. The training procedure used a seed of 0, batch size of 128, over 4000 epochs, with the EDM loss function configured with $p_{\text{mean}} = -1.2$, $p_{\text{std}} = 1.2$, one noise sample per iteration, and a minimum sigma of 0.

**Fine-tuning**  For fine-tuning on the STL-10 dataset, the learning rate was increased to $3 \times 10^{-4}$, while continuing with the Adam optimizer and no weight decay. The batch size was set to 19,384, the number of epochs was reduced to 50, and the number of noise samples was increased to 8. Differential privacy parameters included $\alpha_{\text{num}} = 100$, $\alpha_{\min} = 500$, $\alpha_{\max} = 1500$, a maximum gradient norm of 0.001, $\delta = 1 \times 10^{-5}$, and $\epsilon = 0.99$. The data was divided into 128 splits to facilitate memory-efficient training.

#### B.1.3  DPDM (DOCKHORN ET AL. (2023))

The DPDM method involved training the pretrained NCSN++ model on the STL-10 dataset using the Adam optimizer with a learning rate of $3 \times 10^{-4}$ and no weight decay. The batch size was set to 128, and the model was trained for 50 epochs. A deterministic DDIM sampler with 500 steps was used, with a time range from $t_{\min} = 0.002$ to $t_{\max} = 80$, and $\rho = 7$. To optimize the privacy-utility trade-off, the number of noise samples per iteration was set to 8. The differential privacy parameters included $\alpha_{\text{num}} = 100$, $\alpha_{\min} = 500$, $\alpha_{\max} = 1500$, a maximum gradient norm of 0.001, $\delta = 1 \times 10^{-5}$, and $\epsilon = 1.00$. The dataset was split into 128 parts to optimize memory usage during training.

### B.2  CIFAR-10 (COATES ET AL. (2011))

#### B.2.1  DPSDA (LIN ET AL.)

Images were generated using class-conditioned DPSDA with a pretrained improved diffusion model on ImageNet at a resolution of $32 \times 32$. InceptionV3 (Szegedy et al. (2016)) was used as the feature

---

[1]https://github.com/microsoft/DPSDA

[2]https://github.com/nv-tlabs/DPDM

[3]https://github.com/SunnierLee/DP-ImaGen

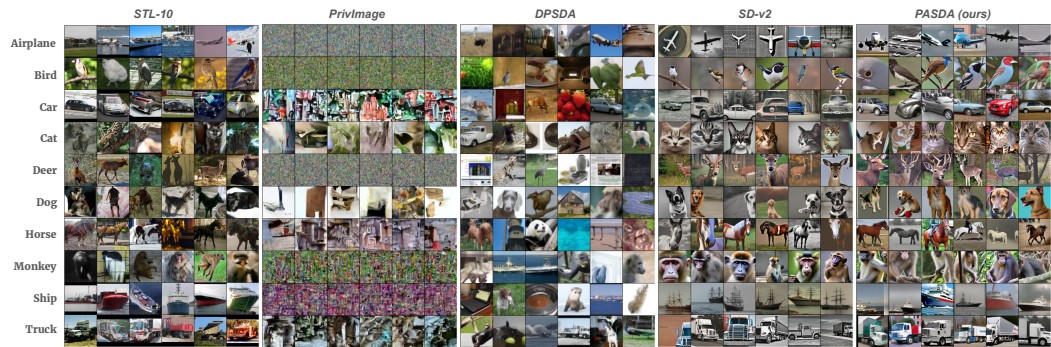

Figure 7: Comparison of original STL-10 images (leftmost column) with synthetic images generated by various approaches (the four columns to the right).

extractor, with a count threshold of 10 and a lookahead degree of 1. The noise multiplier was calculated as 15.83 under the privacy parameters $\epsilon = 1$ and $\delta = 10^{-5}$. The procedure involved 20 iterations, with 50,000 samples generated per iteration, and a linear increase in the degree of variation from 0 to 40 over the iterations.

### B.2.2 PRIVIMAGE (LI ET AL. (2024))

**Semantic Query** The semantic query model remained the same as used for STL-10 (section B.1.2).

**Pretraining** Pretraining followed the same procedure as outlined for STL-10, but at a resolution of $32 \times 32$. The NCSN++ model was trained using the EDM framework, with attention at a resolution of 16, and channel multipliers of $[1, 2, 4]$. Training was conducted over 4000 epochs with a batch size of 512.

**Fine-tuning** For fine-tuning on CIFAR-10, the optimizer's learning rate was increased to $3 \times 10^{-4}$, with the Adam optimizer and no weight decay. The batch size was set to 19,384, and the number of noise samples increased to 8. Differential privacy parameters included $\alpha_{num} = 100$, $\alpha_{min} = 500$, $\alpha_{max} = 1500$, a maximum gradient norm of 0.001, $\delta = 1 \times 10^{-5}$, and $\epsilon = 0.99$. The dataset was partitioned into 128 splits to manage memory usage.

### B.2.3 DPDM (DOCKHORN ET AL. (2023))

For DPDM on CIFAR-10, the pretrained NCSN++ model was trained with a batch size of 2048 and a learning rate of $3 \times 10^{-4}$, using the Adam optimizer without weight decay. The model was trained for 50 epochs, with a deterministic DDIM sampler utilizing 500 steps and the same privacy settings as described for STL-10. The dataset was similarly partitioned into 128 parts for memory efficiency.

## C MORE SAMPLE IMAGES

We further provide visualizations of the synthetic datasets generated by various methods on STL-10. Due to the increased challenge posed by STL-10, characterized by its high resolution and small dataset size, PrivImage fails to produce semantically meaningful images, with most outputs resembling random noise. While DPSDA generates visually coherent images, it struggles to accurately match the generated images with the correct categories. In contrast, PASDA consistently produces high-quality images with a distribution closely aligned with that of STL-10.

