# OpenReview forum: "Advancing Differential Privacy through Synthetic Dataset Alignment"
_ICLR.cc/2025/Conference — ICLR 2025 Conference Withdrawn Submission_

### Official Review · Reviewer_8TR2 · 2024-11-01

**Soundness:** 2
**Presentation:** 3
**Contribution:** 2
**Rating:** 1
**Confidence:** 5

**Summary:**

This paper proposes a novel technique for generating differentially private synthetic data. To minimize the utility loss induced by the noise introduced for differential privacy, the authors leverage a pre-trained model (stable diffusion) and use only the statistics computed on the private dataset, which minimizes access to private data. Additionally, this approach is efficient, as it does not require fine-tuning the pre-trained model and does not require iterative optimization of the synthetic data.

**Strengths:**

This paper addresses the well-known privacy-utility tradeoff and proposes an approach that effectively optimizes this balance. Additionally, the approach is efficient as it does not require iterative optimization of synthetic data or fine-tuning of the pre-trained model.

**Weaknesses:**

W1: Unless I am mistaken, the concept of a neighboring dataset in this paper is not clearly defined. However, given that the full data set for each class is used each time, I assume the subsampling method employed is without-replacement subsampling, consistent with the replace-one approach. Consequently, the sensitivity that should be applied here is $2\kappa$ rather than $\kappa$.

W2: I have identified several errors and typos in the paper. Here are a few examples:

- "Gaussian noise to the domain gap vector" : Gaussian noise is added to the aggregated embedding vector.
- "$\hat{\Delta}v=\Delta v + \mathcal{N}(0,\sigma^2)$" : same remark as in the previous point + $\kappa$ is not mentioned.
- In Algorithm 2, line 10, $s$ is used instead of $\kappa$. Therefore, I do not see the point in defining $\kappa$ if it is not utilized.

**Questions:**

Could you please clarify the points mentioned in W1, as they are crucial for asserting the correctness of the DP protection?

---

### Official Review · Reviewer_W7dH · 2024-11-03

**Soundness:** 3
**Presentation:** 3
**Contribution:** 3
**Rating:** 3
**Confidence:** 4

**Summary:**

The paper investigates the current problem of generating high utility differentially private synthetic datasets. Particularly, apart from the previous approaches of either training a differentially private generative model or generating the synthetic datasets iteratively via the foundational APIs, the paper proposes PASDA where only two steps are required to create the differentially private synthetic data. Extensive experiments showed the effectiveness of the proposed method.

**Strengths:**

1. The proposed technique is novel as it incorporates both the idea of one-step consumption of budgets in the DP generative model trend and the idea of using large foundational APIs as in DPSDA.
2. The main experiments and ablation studies are extensive and explain certain reason why PASDA would work.

**Weaknesses:**

1. It seems to me that PASDA here prior to calculating the domain gap performs a partition based on the category of the training data here which doesn't seems to be satisfying the full definition of differential privacy here (where the definition gives guarantees on both the data and label). On the other hand, doing Hungarian matching may also exploit or leak the information of private data which should be taken care of. I would suggest the authors to provide a full proof of privacy concerning the details of PASDA, instead of only the conversion of RDP and DP with typical Gaussian Mechanism.
2. The experiments should also resolve some concerns mentioned in the Privacy Concerns in Pre-training Data by using data that are certainly not included in most pre-training data. For instance, DPSDA [1] proposes an alternate benchmark of cats that has never been uploaded to the Internet. It would be great if the authors could also experimented in a similar fashion to see the features that PASDA could capture.
3. More experiments on the performance comparison against similar methods could be carried out. For instance, [2] and [3] uncover the connection of DP and dataset condensation (DC) which is similar to PASDA's fashion in building synthetic datasets by means of the original information of private data.

[1]: Zinan Lin, Sivakanth Gopi, Janardhan Kulkarni, Harsha Nori, and Sergey Yekhanin. Differentially private synthetic data via foundation model apis 1: Images. In ICLR. \
[2]: Zheng, Tianhang, and Baochun Li. "Differentially Private Dataset Condensation." In NDSS.\
[3]: Vinaroz, Margarita, and Mijung Park. "Differentially Private Kernel Inducing Points using features from ScatterNets (DP-KIP-ScatterNet) for Privacy Preserving Data Distillation." Transactions on Machine Learning Research.

**Questions:**

1. The authors should clarify here either PASDA stands for Privacy-Aware Synthetic Dataset Alignment or privatePrivacy-Aware Synthetic Dataset Alignment as both appears in the article and the upper and lower cases are also used in a mixed fashion.

---

### Official Review · Reviewer_eZwo · 2024-11-04

**Soundness:** 1
**Presentation:** 3
**Contribution:** 1
**Rating:** 1
**Confidence:** 5

**Summary:**

The paper proposes a method called Privacy-Aware Synthetic Dataset Alignment (PASDA) to generate differentially private synthetic data for downstream model training. First, PASDA generates a fully synthetic dataset using a pre-trained class-conditional generative model (e.g., Stable Diffusion) without accessing the private data. Second, it aligns the synthetic data's distribution with the private data by extracting differentially private feature statistics from the private dataset and using them to adjust the synthetic data's embeddings in CLIP space. This adjusted embedding is then decoded back into image space using a pre-trained decoder (unCLIP).

**Strengths:**

* Important problem
* Promising idea on how to use pre-trained foundation models for the task

**Weaknesses:**

Despite what the paper claims, as presented PASDA does not offer differential privacy guarantees. To understand why we can look at a simplified version of the algorithm: the mechanism gets a dataset of feature vectors (i.e. CLIP embeddings of private images), clusters the vectors into K clusters and releases the centroids of each cluster with noise calibrated to hide the presence of a single vector in the cluster. The critical problem is that the way the noise is added does not account for sensitivity of the clustering function: *if* one could guarantee that changing one point in the dataset would keep the cluster assignments of all other points exactly the same, then at most two (or one, depending on the neighbouring relation) of the centroids would change by the contribution of one point and the present analysis would be correct. Unfortunately, this is not necessarily the case: changing one point can completely change the clustering inducing much higher sensitivity on the cluster centroids than claimed in the paper.

To see this consider a one-dimensional dataset D_0 constructed as follows: pick a large enough constant M > 0, take the intervals [0,1], [M, M+1], [2M, 2M+1] and sample 10K points uniformly within each of the intervals. Now take D = D_0 U {(M+1)/2} and D' = D_0 U {(M+1 + 2M)/2}: in D we add a point between the first and second interval and in D' between the second and third interval. When asked to produce 2 clusters, any reasonable clustering algorithm will cluster D as C_1 = {all points between 0 and M+1} and C_2 = {all the points between 2M and 2M+1}, while on D' it will produce clusters C_1' = {all points between 0 and 1} and C_2' = {all points between M and 2M+1}. It is clear from this example that one can make the sensitivity of the cluster centroids arbitrarily large. Note that clipping can mitigate some of this effect, but would still result in a sensitivity that scales with the number of points.

**Questions:**

* Have you considered making the clustering and matching step differentially private? Note that DP clustering is a notoriously hard problem, but some practical algorithms exist in the literature.

---

### Official Review · Reviewer_zMcy · 2024-11-04

**Soundness:** 2
**Presentation:** 3
**Contribution:** 2
**Rating:** 5
**Confidence:** 4

**Summary:**

This paper explores differentially private (DP) data synthesis using pre-trained models. The proposed method trains synthetic data, initialized from samples generated by pre-trained conditional diffusion models, to match the CLIP embeddings of real data, then uses UNCLIP to update the synthetic data from these embeddings. Differential privacy is incorporated through DP mean computation of the CLIP embeddings. Experimental results demonstrate promising downstream utility of this approach.

**Strengths:**

- The paper addresses a valid and important topic, and it’s generally well written
- The proposed method is straightforward to implement and readily applicable in practice.
- Experimental results are promising, highlighting the potential of using pre-trained models for DP data generation.

**Weaknesses:**

- The privacy computation lacks clarity: there is no explicit calculation of the L2-sensitivity for this specific case, nor is the privacy notion (e.g., add-one/replace-one) clearly defined. Additionally, the number of clusters $K$ and the number of label categories $c$ should have a direct linear impact on the resulting privacy cost $\epsilon$. This influence, however, is neither reflected in the pseudocode nor in Corollary 1, which raises significant concerns regarding the validity of the claimed privacy guarantees. Without a clear and justified privacy computation, it is difficult to properly assess the contribution.
- The clarity of the algorithm presentation (and some experimental results) could be further enhanced to improve overall readability and understanding (see below for details).
- The use of feature matching for DP data generation is a well-established approach, which diminishes the technical novelty of this paper. Prior works, such as the following, have already explored this concept:
   - “Dp-merf: Differentially private mean embeddings with randomfeatures for practical privacy-preserving data generation”, AISTAT 2021
  - “Differentially private sliced wasserstein distance. In International Conference on Machine Learning”,  ICML 2021
  - “Pearl: Data synthesis via private embeddings and adversarial reconstruction learning”, ICLR 2022
  - “Hermite polynomial features for private data generation.”, ICML 2022
  - “Differentially private neural tangent kernels for privacy-preserving data generation.”, Journal of Artificial Intelligence Research

**Questions:**

-  As noted above, a clear explanation of the privacy cost computation is necessary, particularly to address why the factors of the number of clusters and categories are not reflected in the privacy cost.
- The definition of $\mathcal{C}_m^{\text{syn}}$ is not presented in Algorithm 1, which reduces clarity.
-  It would be beneficial to explicitly include the subscripts for $(\mathcal{S}^{\text{(syn)}}, \mathcal{S}^{(priv))}$ in Line 11 of Algorithm 1 to improve readability
- The authors should prominently specify the pre-training datasets used for the pre-trained models (e.g., diffusion, CLIP, UNCLIP) across all methods, including baselines and the proposed approach, to ensure better comparison.  There is concern that improvements in the proposed method could be merely attributed to stronger pre-trained models or larger pre-training datasets.
-A comparison of methods from Table 1 under varying $\epsilon$ values (e.g., plotting each method as a curve with utility metric on the y-axis and $\epsilon$ on the x-axis) would provide clearer and fairer insights into each method’s performance.
- Robustness to large domain shifts between private and public pre-training data is a critical aspect when using pre-trained models, yet this paper does not fully investigate this factor. Improvement could be made by testing on data with specific domain features, such as medical data.
- The results in Table 1 indicate poor performance of DPSDA on the CIFAR-10 dataset (25.7%, 47.1% test accuracy), which seems inconsistent with the performance reported in the original paper (Figure 5 shows DPSDA achieving >85% test accuracy under $\epsilon = 3.34$). The discrepancy here is confusing and warrants clarification.

---

### Note · Authors · 2024-11-20

I have read and agree with the venue's withdrawal policy on behalf of myself and my co-authors.